# Spectra Stable Quantum Dots Enabled by Band Engineering for Boosting Electroluminescence in Devices

**DOI:** 10.3390/mi13081315

**Published:** 2022-08-14

**Authors:** Bingbing Lyu, Junxia Hu, Yani Chen, Zhiwei Ma

**Affiliations:** 1School of Physics, Harbin Institute of Technology, Harbin 150001, China; 2School of Information Engineering, Xinyang Agriculture and Forestry University, Xinyang 464000, China; 3Institute for Advanced Study, Shenzhen University, Shenzhen 518060, China; 4Center of Materials Science and Optoelectronics Engineering, University of Chinese Academy of Sciences, Beijing 100049, China

**Keywords:** quantum dots, band engineering, charge injection balance, electroluminescent devices

## Abstract

The band level landscape in quantum dots is of great significance toward achieving stable and efficient electroluminescent devices. A series of quantum dots with specific emission and band structure of the intermediate layer is designed, including rich CdS (R-CdS), thick ZnSe (T-ZnSe), thin ZnSe (t-ZnSe) and ZnCdS (R-ZnCdS) intermediate alloy shell layers. These quantum dots in QLEDs show superior performance, including maximum current efficiency, external quantum efficiencies and a T_50_ lifetime (at 1000 cd/m^2^) of 47.2 cd/A, 11.2% and 504 h for R-CdS; 61.6 cd/A, 14.7% and 612 h for t-ZnSe; 70.5 cd/A, 16.8% and 924 h for T-ZnSe; and 82.0 cd/A, 19.6% and 1104 h for R-ZnCdS. Among them, the quantum dots with the ZnCdS interlayer exhibit deep electron confinement and shallow hole confinement capabilities, which facilitate the efficient injection and radiative recombination of carriers into the emitting layer. Furthermore, the optimal devices show a superior T_50_ lifetime of more than 1000 h. The proposed novel methodology of quantum dot band engineering is expected to start a new way for further enhancing QLED exploration.

## 1. Introduction

Colloidal quantum dots possess unique optical properties, including a narrow emission spectrum, high photoluminescence efficiency, a continuously tunable spectral range, solution processability and other excellent optical properties [1,2,3,4,5,6]. All these attractive properties of quantum dots make them excellent candidates for the development of next-generation display technologies. The numerous advantages of quantum dot light-emitting diodes (QLEDs) in terms of color purity, stability and production cost may provide a promising solution for next-generation wide-gamut color displays and lighting technologies [4,7,8,9,10,11,12,13].

In recent years, the in-depth understanding of the physical mechanism of QLEDs, especially based on quantum dot nanostructure tailoring to limit their efficiency and stability, has greatly improved device performance [14,15,16,17,18,19,20,21]. Qian et al. used quantum dot nanostructure tailoring to achieve high-efficiency QLEDs and obtained an external quantum efficiency (EQE) of over 10% and a half lifetime of a green device of over 90,000 h [9]. Jang et al. utilized these methods, and not only was a EQE of over 20% achieved, but also the half lifetime of a red QLED device reached over 100,000 h [13]. However, for the fine-tuning of quantum dot nanostructures in specific emission wavelength ranges [22,23], the band structure of the continuously grown shell of quantum dots when regulating the charge transport and radiative recombination efficiency in devices remains challenging. It has been demonstrated that excitons induce the spatial separation of electrons and holes in weak quantum dot electron confinement under the action of an electric field or photoexcitation, thereby reducing electron–hole overlap and resulting in reduced radiative recombination efficiency [8,9,24,25]. This result further leads to exciton quenching by trapping and delocalizing on the surface of quantum dots and auger recombination when the device is charged at a high driving current, resulting in a rapid roll-off of EQE. At the same time, strong quantum confinement in quantum dots is not conducive to balancing the charge injection of electroluminescent devices and limit the performance of QLEDs.

Here, we propose quantum dot band engineering for efficient and stable electroluminescent devices. Specifically, we designed a series of quantum dots with intermediate transition shells with different quantum confinement capabilities. The intermediate transition layers of green CdSe@ZnS alloy quantum dots include Rich-CdS, Rich-ZnSe and Rich-ZnCdS, which balance the electron and hole injection efficiency and improve device performance. Among them, the Rich-ZnCdS quantum dots as the emitting layer show a peak EQE of up to 19.6%, suppression efficiency roll-off characteristics (EQE > 15% between 100 and 10,000 cd/m^2^) and long-term operating stability (T_50_ at more than 1000 h at a brightness of 1000 cd/m^2^). The superior performance and low-cost manufacturing potential of QLEDs is a critical step toward commercializing display technology.

## 2. Materials and Methods

### 2.1. Synthesis of Quantum Dots by Band Engineering

Typically, the CdSe@ZnS quantum dots with a rich CdS middle shell were prepared by the following method [26]. 0.2 mmol of cadmium oxide and 3.9 mmol of zinc acetate (Zn(oAc)_2_) were placed with 5 mL of oleic acid (OA) in a 100 mL flask. Then, 15 mL of octadecene was injected into the reaction flask and heated to 170 °C; it was stirred for 20 min under the nitrogen flow, yielding a clear mixture solution of Cd(OA)_2_ and Zn(OA)_2_. Then, the mixture was heated up to 300 °C for 30 min. Afterwards, 0.2 mmol of Se and 3.8 mmol of S were dissolved in 2 mL of trioctylphosphine (TOP) and swiftly injected into the mixture solution and kept at 300 °C for 10 min to form the CdSe@ZnS quantum dots with a rich CdS middle shell. Next, the reactor was lowered to room temperature. The purification procedures were performed by using the dispersion/precipitation method with toluene/ethanol, and the procedure was repeated 5 times. The prepared CdSe@ZnS quantum dots were then dispersed in toluene after post-treatment by our previously reported method [26] and were ready for further usage.

### 2.2. Synthesis of Zinc Oxide Nanocrystals (ZnO NPs)

ZnO NPs were synthesized by the reported method [9]. 3 mmol of Zn(oAc)_2_ dissolved in 30 mL of dimethyl sulphoxide and 5.5 mmol of tetramethylammonium hydroxide pentahydrate dissolved in 10 mL of ethanol were mixed and stirred for 1 h at 23 °C. Then, the ZnO NPs were washed with ethanol/ethyl acetate (1/4, *v*/*v*) and centrifuged (8000 rpm, 3 min) twice. Finally, the ZnO NPs were dispersed in ethanol (30 mg/mL) and stored at −4 °C for usage.

### 2.3. Device Fabrication

The QLEDs were fabricated by spin coating on patterned ITO glass substrates (sheet resistance ∼18 Ω/sq). The substrates were cleaned in ultrasonic baths of detergent, deionized water, chromatographic grade acetone and 2-propanol for 15 min each, and the cleaned ITO was exposed to a UV-ozone treatment for 15 min. Then, the poly (ethylenedioxythiophene) and polystyrene sulphonate (PEDOT: PSS, filtered with a 0.45 μm PVDF filter) were spin-coated on the substrates at 40 nm thickness at a spin rate of 4000 rpm for 45 s and baked at 140 °C for 15 min in air. Next, they were then transferred into an N_2_-filled glove box for further spin coatings of poly [9], 9-dioctyl-fluorene-co-N-(4-butylphenyl)-diphenylamine] (TFB), quantum dots and ZnO NPs layers. Afterwards, TFB, the quantum dot solution and the ZnO NPs, filtered with a 0.22 μm PVDF filter before use, were then spin-coated onto the TFB (8 mg/mL, in chlorobenzene) layer at 2000 rpm for 45 s, followed by thermal annealing at 150 °C for 30 min. The quantum dots were spin-coated on the ITO/PEDOT: PSS/TFB layer, and the optimized emission layer thicknesses were ∼30 nm (15 mg/mL, 2500 rpm for 45 s); the ZnO NPs layer was spin-coated on the ITO/PEDOT: PSS/TFB/quantum dots layer and then baked at 60 °C for 30 min. Finally, the multilayered device samples were loaded into a high-vacuum chamber (∼1 × 10^−7^ torr) for deposition of an Al cathode (100 nm). To protect the devices from water and oxygen, all the devices were encapsulated with UV-cured epoxy and covered with thin glass slices.

### 2.4. Characterization

Steady-state PL spectra, quantum yields and time-correlated single-photon counting spectrofluorometer were collected using an Edinburgh FLS920 fluorescence spectrophotometer. PL decay spectra of quantum dot samples were excited by a 405 nm ps laser. The PL spectral stability was performed using a commercial UV-LED with a corresponding emission wavelength of 365 nm and a power of 5 W. Device performance was tested at room temperature with an ambient humidity of less than 50%. The size of the quantum dots was obtained by a JEOL JEM-2100F transmission electron microscope (TEM) at 200 KV. The post-process details are the following: the quantum dot samples were prepared in a toluene solution and all the labeled samples were steadily shaken to mix the samples evenly. Then, during the treatment process, all the samples were treated with the same shock during the processing period, and the samples were labeled and kept under UV lamp irradiation for 10 h until the end of the process. (Note: the spherical UV lamp light wavelength was ∼365 nm, the power was ∼5 W and the light flux was ∼350 lm). The current-voltage-luminance characteristics of the QLEDs were measured under ambient conditions, the electroluminescence spectra and luminance were obtained by a PR-735 spectroradiometer (Photo Research) and a Keithley 2400 source meter was used to furnish the driving voltage and to record its current density.

## 3. Results and Discussion

Based on the fabrication strategy to tame quantum dot nucleation and growth, we further modified the method to create nanostructures with a controllable graded alloy intermediate shell sandwiched between a CdSe-rich inner core and ZnS-rich outer shells. Rich CdS (R-CdS), thick ZnSe (T-ZnSe), thin ZnSe (t-ZnSe) and ZnCdS (R-ZnCdS) intermediate alloy shell layers were prepared by finely designing the relative ratios and reaction concentrations of the anion and cation precursors, and the continuously graded alloy structure formed quantum dots with different wave function confinement capabilities for the luminescent core (Figure 1a). Hereafter, a series of green-emitting CdSe@ZnS quantum dots with different interlayer structures were synthesized. In detail, the molar ratios of the intermediate shell quantum dots are Zn/Cd (3.9/0.3) and TOP-Se/S (0.2/3.8, 2 mL) for R-CdS, Zn/Cd (4/0.2) and TOP-Se/S (0.3/3.9, 1.9 mL) for t-ZnSe, Zn/Cd (4.05/0.15) and TOP-Se/S (0.4/3.8, 1.8 mL) for T-ZnSe and Zn/Cd (3.8/0.4) and TOP-Se/S (0.15/4.05, 1.8 mL) for R-ZnCdS. Compared with R-CdS quantum dots, the conduction band energy level position of the ZnSe-rich or ZnCdS-rich quantum dots is increased by 0.7 eV, and the valence band position is decreased by 0.2 eV, providing a better exciton wave function confinement capability [27]. Theoretically, these results are beneficial for reducing the delocalization of excitons to the quantum dots surface due to the weak confinement of the electron wave function, thereby effectively suppressing the nonradiative auger recombination process [28,29,30,31].

Figure 1b shows the absorption and fluorescence spectra of green quantum dots enriched with CdS, ZnSe and ZnCdS in the intermediates transition shell. Although they have different nanostructures and band offsets, their characteristic absorption and emission peaks are almost identical (Figure 1b). The different intermediate layer types have very analogous PL emission peaks at λ = 532 nm (T-ZnSe), 531 nm (t-ZnSe), 533 nm (R-CdS) and 532 nm (R-ZnCdS). Moreover, CdS-rich thick-shell ZnSe, thin-shell ZnSe and ZnCdS quantum dots have similar near-unity PL quantum yields (>90%) and PL decay kinetics (Figure 1c). Compared with the T-ZnSe, t-ZnSe and R-CdS quantum dots, the R-ZnCdS quantum dots have the smallest non-radiative recombination lifetime and proportion (Table 1). Since the middle shell of quantum dots better matches the crystal structure from the inner core to the outer shell, it reduces lattice defects and thus obtains a relatively high performance, resulting in excellent radiative recombination efficiency. Figure 2 presents the transmission electron microscope (TEM) images of the T-ZnSe, t-ZnSe, R-CdS and R-ZnCdS quantum dots produced by band engineering. These quantum dots have average sizes of 9.5 nm for R-CdS, 9.6 nm for T-ZnSe, 9.7 nm for t-ZnSe and 9.8 nm for R-ZnCdS, all of which are well maintained a spherical shape with a narrow size distribution (Figure 2a–d).

To further investigate the effect of band engineering on the spectra stability of quantum dots, the above R-CdS, T-ZnSe, t-ZnSe and R-ZnCdS quantum dots were continuously irradiated using a commercial 365 nm UV LED chip, as shown in Figure 3a. The quantum dot fluorescence decay was characterized, and the intensity over time was evaluated by measuring their PL intensity under constant power density illumination [32,33]. The results show that under the same initial conditions, the PL intensity of T-ZnSe quantum dots decreased to 50% of the initial value when the acceleration aging time reached 200 h, while that of t-ZnSe quantum dots is reduced to 62%. Importantly, the R-CdS and R-ZnCdS quantum dot PL intensity decreased by only 5% and 9% after 200 h of continuous irradiation, respectively. Under continuous strong light irradiation, the quantum dots enriched in the ZnSe intermediate transition layer would cause selenium to be oxidized, resulting in a decrease in its PL quantum yield; however, R-CdS and R-ZnCdS quantum dots are resistant to photo-oxidation and exhibit excellent PL stability. At the same time, the results of the quantum dot PL spectra of the accelerated aging show that R-CdS quantum dots exhibited a large spectral redshift (13 nm) under continuous UV light irradiation (Figure 3b). However, the PL profile of T-ZnSe, t-ZnSe and R-ZnCdS quantum dots remained unchanged, keeping the same as the initial PL spectra with only the intensity declining. This result implies that the stability of R-ZnCdS quantum dots is expected to meet the performance requirements of light-emitting devices.

The transport properties of charge carriers across heterostructures have a significant impact on the performance of electronic devices using quantum dots. The electron-only and hole-only devices can better study the transport properties of charge carriers in QLEDs. The charge transport properties of band engineering quantum dots were constructed as single-hole and single-electron devices using ITO/PEDOT: PSS/TFB/quantum dots/TFB/MoO_3_/Al (Figure 4a inset) and ITO/ZnO/quantum dots/ZnO/MoO_3_/Al structure (Figure 3b inset) stacked structure, respectively [29]. The R-CdS and t-ZnSe quantum dots in single-hole devices limit their in-device hole transport efficiency due to the strong hole confinement ability (Figure 4a). As a comparison, R-ZnCdS quantum dots exhibited excellent hole transport properties and hole current density greater than all other quantum dot-based devices, although the hole transport properties of T-ZnSe quantum dots were improved with increasing the driving voltage. As the voltage increases, the charge flow through the device is blocked by the hole barrier of the quantum dot shell structure. When testing single-hole devices, the thick ZnSe middle layer has a shallow hole confinement effect, and the outermost ZnS shell thickness is significantly smaller than that of t-ZnSe and R-CdS. The current density increases rapidly at high voltages (between 3 V and 4 V) that are close to the bandgap region of the ZnSe bulk semiconductor.

It is particularly clear that the change of the valence band position of the quantum dot transition layer structure (0.2 eV) regulates the hole transport performance of the quantum dots in a device, and the change value of the conduction band position is more obvious than the change of the valence band position (0.7 eV), which inevitably leads to the opposite of the current transport characteristics of single-electron devices [34]. Meanwhile, R-CdS quantum dots have higher electron transport properties (Figure 4b), and R-ZnCdS has lower electron transport properties, which is consistent with the band engineering induced energy level structure changes. The R-ZnCdS quantum dots deep electron confinement and shallow hole confinement of quantum dots are more beneficial to balance electron and hole injection into devices, which would be significant for improving the performance of QLEDs.

The device structure of the solution processed QLEDs is schematically shown in Figure 5a and the band-engineered quantum dots used as emission layers. The stacked architecture consists of PEDOT: PSS, TFB and ZnO nanoparticles as the hole injection/transfer layer and the electron transfer layer. The QLEDs were fabricated with the same processes as our previous work [26]. The current density and luminance versus voltage characteristics of QLEDs based on the above four green quantum dots with a different rich intermediate shell are shown in Figure 5b. The R-ZnCdS quantum dot-based device exhibits much lower leakage of current and drive current than a T-ZnSe, t-ZnSe and R-CdS quantum dot device (at V < 3 V). When the driving voltage is greater than 3 V, the injection current density of the device increases rapidly under the operating current, thereby improving the radiative recombination efficiency of the device. This can be attributed to the shallow interlayer valence band offset of T-ZnSe and R-ZnCdS quantum dots, which can better inject holes into the emitting layer. The turn-on voltage of these devices is as low as 2.4 V for T-ZnSe and R-ZnCdS quantum dots, and 2.6 V for t-ZnSe and R-CdS quantum dot devices, consistent with their band gap.

The current efficiency and EQE as a function of luminance for four green QLEDs are shown in Figure 5c. The four green devices exhibit a maximum luminance and current efficiency of 81,200 cd/m^2^ and 47.2 cd/A for R-CdS QLEDs, 129,260 cd/m^2^ and 61.6 cd/A for t-ZnSe devices, 103,410 cd/m^2^ and 70.5 cd/A for T-ZnSe-rich devices and 112,290 cd/m^2^ and 82.0 cd/A for R-ZnCdS devices, achieving a corresponding maximum EQE up to 11.2%, 14.7%, 16.8% and 19.6%, respectively (Table 2). Notably, the brightness of R-ZnCdS-based QLEDs is between 100 and 10,000 cd/m^2^, the driving voltages are only 3 to 4 V and the EQE is always greater than 15%, indicating efficient injection and radiative recombination of holes and electrons into the emitting layer at low voltages.

To evaluate the stability of band engineered quantum dots in LEDs, all devices’ operating lifetime measurements were performed in air at a constant drive current with an initial luminance of 1000 cd/m^2^ (Figure 5d). The brightness slowly decreases from the initial value to half after 504, 612, 924 and 1104 h for T-ZnSe, t-ZnSe, R-CdS and R-ZnCdS quantum dot-based devices, respectively. This can be attributed to the deep conduction band and shallow conduct band offset enabling better electron and hole transport and radiative recommunication in QLEDs, which is beneficial to reducing the thermal quenching of the emitting layer and facilitating the stability of the device.

## 4. Conclusions

In summary, we have demonstrated the intermediate shell band level of quantum dots is of great importance toward realizing high performance QLEDs. The detailed ZnSe, CdS and ZnCdS intermediate shells of quantum dots, especially the deep electron confinement and shallow hole confinement capabilities which effectively balance charge injection/transport and enhance radiative recombination, were found to be essentially important for boosting electroluminescent devices. Stable and reliable R-ZnCdS quantum dots not only exhibit high spectral stability under UV irradiation, but also maximize the performance of QLEDs, including maximum EQEs of up to 19.6%, device lifetimes that surpass 1000 h with an initial brightness of 1000 cd/m^2^ and low operating voltages. We believe that the present quantum dot band engineering will be an important step for the realization of high-performance display and lighting applications.

## Figures and Tables

**Figure 1 micromachines-13-01315-f001:**
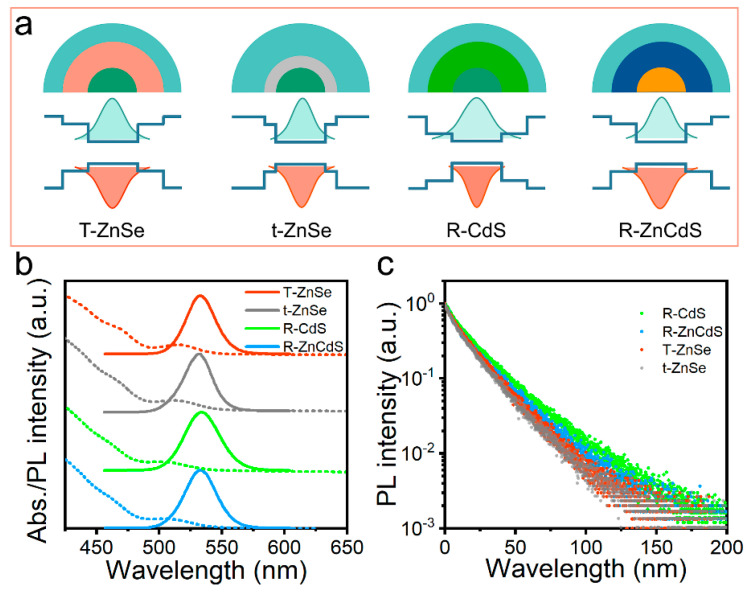
(**a**) Schematic illustration of the nanostructure and wave function confinement properties of rich CdS (R-CdS), thick ZnSe (T-ZnSe), thin ZnSe (t-ZnSe) and ZnCdS (R-ZnCdS) intermediate transition shell quantum dots. (**b**) Photoluminescence (PL) spectra and (**c**) PL decay dynamics of R-CdS, T-ZnSe, t-ZnSe, and R-ZnCdS corresponding quantum dots, respectively.

**Figure 2 micromachines-13-01315-f002:**
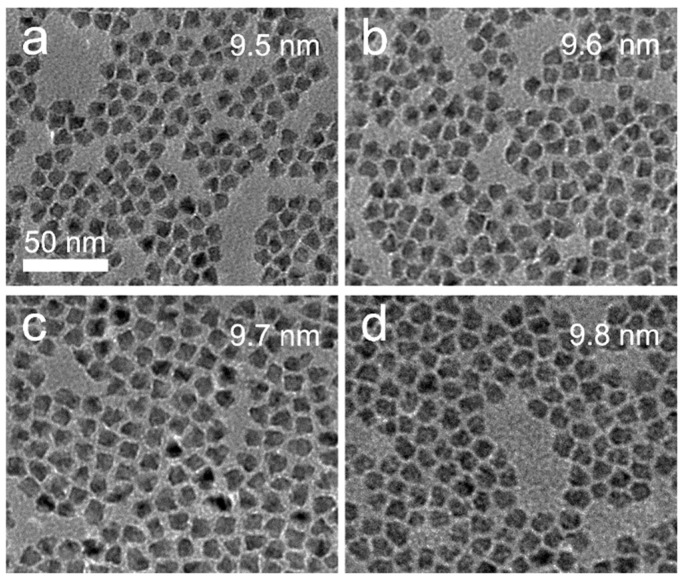
Transmission electron microscope images of the (**a**) R-CdS, (**b**) T-ZnSe, (**c**) t-ZnSe and (**d**) R-ZnCdS quantum dots.

**Figure 3 micromachines-13-01315-f003:**
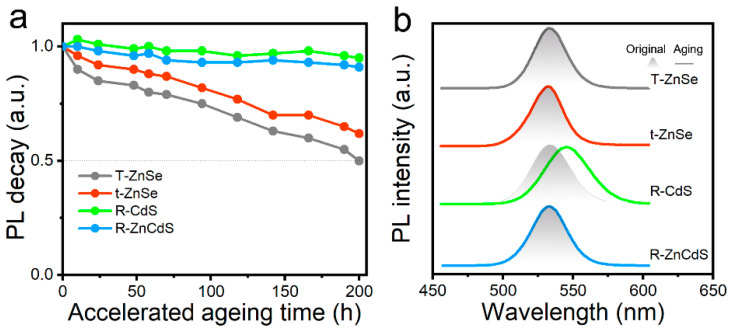
(**a**) PL intensity decay of R-CdS, T-ZnSe, t-ZnSe and R-ZnCdS quantum dots under UV light irradiation. (**b**) Comparison of spectral profiles of accelerated aging quantum dots and original spectra.

**Figure 4 micromachines-13-01315-f004:**
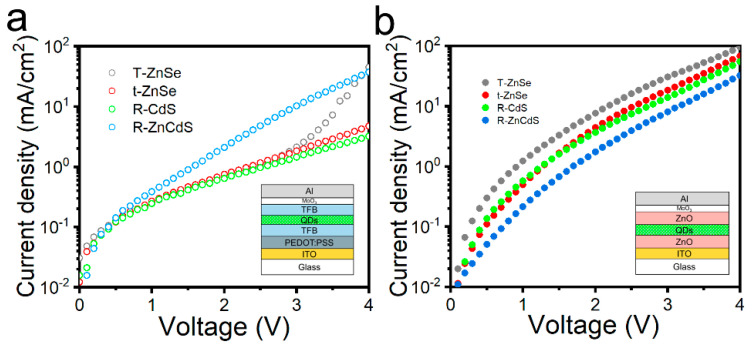
(**a**) Current density-voltage characteristics of hole-only and (**b**) electron-only devices of R-CdS, T-ZnSe, t-ZnSe and R-ZnCdS quantum dots. Inset: schematic diagram of device structure.

**Figure 5 micromachines-13-01315-f005:**
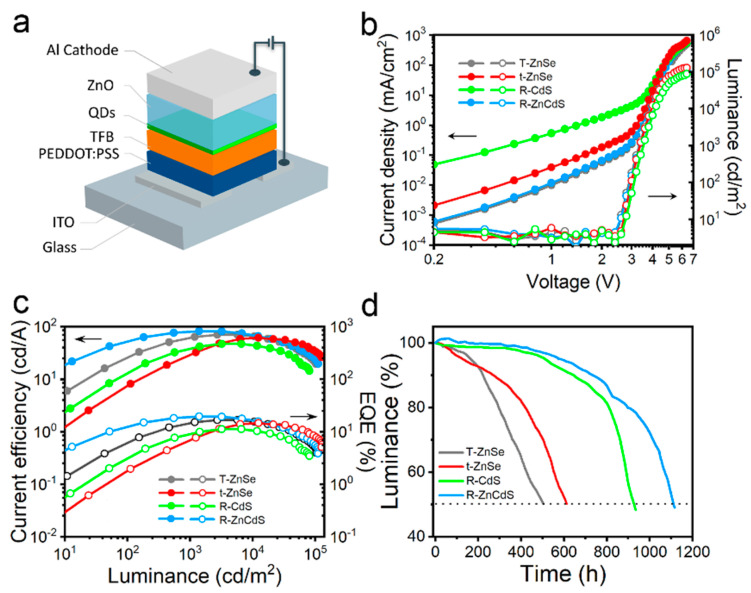
(**a**) Schematic illustration of QLEDs. (**b**) Current density and luminance versus voltage profiles. (**c**) Current efficiency and EQE as a function of luminance of the green QLEDs based on T-ZnSe, t-ZnSe, R-CdS and R-ZnCdS quantum dots. (**d**) Luminance versus time of operation under ambient conditions of T-ZnSe, t-ZnSe, R-CdS and R-ZnCdS devices.

**Table 1 micromachines-13-01315-t001:** PL decay for R-CdS, T-ZnSe, t-ZnSe and R-ZnCdS quantum dots; the nonradiative recombination decay component and its proportion (*τ_1_* and *f_1_*); the radiative recombination component and its proportion (*τ_2_* and *f_2_*); and the corresponding lifetime (*T*).

Structure	*τ_1_* (ns)	*τ_2_* (ns)	*f_1_* (%)	*f_2_* (%)	*T* (ns)
T-ZnSe	4.9	22.9	15.6	84.4	20.1
t-ZnSe	4.7	23.6	18.0	82.0	20.2
R-CdS	6.9	24.5	13.7	86.3	22.1
R-ZnCdS	4.1	23.8	12.9	87.1	21.3

**Table 2 micromachines-13-01315-t002:** Summary of maximum current efficiency (CE, cd/A), luminance (Lmax, cd/m^2^), EQE (%) and lifetime (h) of QLEDs.

Quantum Dots	CE (cd/A)	Lmax (cd/m^2^)	EQE(%)	Lifetime (h)@1000 cd/m^2^
T-ZnSe	70.5	103,410	16.8	504
t-ZnSe	61.6	129,260	14.7	612
R-CdS	47.2	81,200	11.2	924
R-ZnCdS	82.0	112,290	19.6	1104

## Data Availability

Not applicable.

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
