# Peer review of "Spectra Stable Quantum Dots Enabled by Band Engineering for Boosting Electroluminescence in Devices"

_micromachines, 2022, doi:10.3390/mi13081315_

Round 1

Reviewer 1 Report

I recommend acceptance with Minor Revision.

The authors demonstrate a universality strategy to synthesize core-shell QDs with different intermediate layers, and investigate the effect of their interlayer shells on high performance QLEDs. They found that ZnCdS interlayer exhibit deep electron confinement and shallow hole confinement capabilities, which show superior external quantum efficiencies and lifetime 19.6% and more than 1000 hours at 1000 cd/m2 for T50. This work is interesting for publication in Micromachines after Minor Revision and Reinspection.

(1) T50 should be present in the Abstract;

(2) In line 97-98, “The size of the quantum dots obtained by a JEOL JEM-2100F transmission electron microscope (TEM) at 200 KV” should be removed as there is no TEM data.

(3) The full name is needed for the first time the abbreviation appears, such as R-CdZnS in line 107;

(4) There’s no (c) in the caption of Fig. 1;

(5) The power of “commercial 365 nm UV LED chip” in line 141 should be provided.

(6) The authors indicate in line 212 that the QLED device was performed in air. Please indicate whether the device was directly exposed to air or encapsulated with thin glass slices.

Author Response

Dear Reviewer 1:

Thank you very much for your positive and meaningful comments, which will certainly help improve the quality of this manuscript and the core ideas that this manuscript tries to convey.

Comments and details, Please see the attachment. 

Reviewer 2 Report

Minor revision

Author Response

Dear Reviewer 2:

Thank you very much for your positive and meaningful comments, which will certainly help improve the quality of this manuscript and the core ideas that this manuscript tries to convey.

Comments and details, Please see the attachment. 

Reviewer 3 Report

The authors synthesized a series of CdSe@ZnS alloy quantum dots with diffierent nanostructure by band engineering for boosting electroluminescence devices. The optimized quantum dots exhibit controllable energy band nanostructure, which not only elevate the spectral stability of green quantum dots but also further enhances the injection/transportation of charges and improve the radiative recombination efficiency of quantum dots in the device. The final devices show an excellent EQE and long operation lifetime properties. This MS is well written and the results are valuable for QDs nanostructure engineering and robust QLEDs. This MS can be accepted for publishing after addressing the following issues:

Q1. In Fig. 1a, how did the authors get the schematic of QDs energy aligns and exciton delocalization?

Q2. The intermediate transition layers of CdSe@ZnS alloy quantum dots include rich-CdS, Rich-ZnSe and Rich-ZnCdS with an appropriate chemical composition can achieve green emission. The morphology of the corresponding quantum dots should be given.

Q3. For the time-resolved PL decay spectra in Figure 1c, they should provide both original spectra and fitted spectra. In addition, For the recombination types of τ1 and τ2, the corresponding references should be given (J. Phys. Chem. Lett. 2020, 11, 9862-9868).

Q4. In the MS, the QLEDs with Rich-ZnCdS quantum dots exhibit the better injection features of charge into the QDs layer due to the higher valence band edges. And post-processing of quantum dots is equally important, and relevant literature discussion should be added.

Q5 In Figure 3a, a steep rising can be observed between 3 V and 4 V for T-ZnSe. The origin of this rising is suggested to be clearly explained.

Author Response

Dear Reviewer 3:

Thank you very much for your positive and meaningful comments, which will certainly help improve the quality of this manuscript and the core ideas that this manuscript tries to convey.

Comments and details, Please see the attachment. 

Author Response

Dear Reviewer 4:

Thank you very much for your positive and meaningful comments, which will certainly help improve the quality of this manuscript and the core ideas that this manuscript tries to convey.

Comments and details, Please see the attachment. 
